# Role of Omega-3 Fatty Acids as Non-Photic Zeitgebers and Circadian Clock Synchronizers

**DOI:** 10.3390/ijms232012162

**Published:** 2022-10-12

**Authors:** Ana Checa-Ros, Luis D’Marco

**Affiliations:** 1Department of Medicine and Surgery, Faculty of Health Sciences, Universidad Cardenal Herrera—CEU, CEU Universities, 46115 Valencia, Spain; 2Aston Institute of Health and Neurosciences, School of Life & Health Sciences, Aston University, Birmingham B4 7ET, UK; 3Department of Nephrology, Hospital General Universitario de Valencia, 46014 Valencia, Spain

**Keywords:** molecular clock, clock genes, circadian rhythms, melatonin, sleep/wake cycle, fatty acids, omega-3

## Abstract

Omega-3 fatty acids (ω-3 FAs) are well-known for their actions on immune/inflammatory and neurological pathways, functions that are also under circadian clock regulation. The daily photoperiod represents the primary circadian synchronizer (‘zeitgeber’), although diverse studies have pointed towards an influence of dietary FAs on the biological clock. A comprehensive literature review was conducted following predefined selection criteria with the aim of updating the evidence on the molecular mechanisms behind circadian rhythm regulation by ω-3 FAs. We collected preclinical and clinical studies, systematic reviews, and metanalyses focused on the effect of ω-3 FAs on circadian rhythms. Twenty animal (conducted on rodents and piglets) and human trials and one observational study providing evidence on the regulation of neurological, inflammatory/immune, metabolic, reproductive, cardiovascular, and biochemical processes by ω-3 FAs via clock genes were discussed. The evidence suggests that ω-3 FAs may serve as non-photic zeitgebers and prove therapeutically beneficial for circadian disruption-related pathologies. Future work should focus on the role of clock genes as a target for the therapeutic use of ω-3 FAs in inflammatory and neurological disorders, as well as on the bidirectional association between the molecular clock and ω-3 FAs.

## 1. Introduction

### 1.1. The Biological Clock and Its Molecular Machinery

Circadian rhythms refer to the nearly 24 h period associated with the Earth’s rotation cycle under which physiological processes in organisms oscillate [1]. Circadian rhythms reflect the need for diverse organisms, from photosynthetic prokaryotes to higher eukaryotes, to adapt their biological functions to the environment [2]. In this way, for instance, species can regulate their reproductive behavior to favor offspring birth during warmer seasons, when food availability is higher [3].

The synchronization of circadian rhythms requires the alignment of a complex of endogenous oscillators (‘biological clock’) with external stimuli serving as environmental cues, also known as ‘zeitgebers’. In this sense, the daily light/dark cycle represents circadian rhythms’ primary synchronizer (‘zeitgeber’) [4]. However, certain nutrients have been reported to act as non-photic zeitgebers, such as caffeine [5] and dietary polyphenols [6]. The present review aims at providing evidence on the molecular bases of the role of dietary omega-3 (ω-3) fatty acids as non-photic zeitgebers.

The biological clock relies on a highly orchestrated machinery centrally regulated by a dense structure of neurons (around 20,000 in rodents and 50,000 in humans) located in the suprachiasmatic nucleus (SCN) of the hypothalamus [7,8]. The SCN, whose neurons contain cell-autonomous and self-sustained clocks, is considered the circadian master pacemaker, as it acts hierarchically to synchronize multiple peripheral clocks contained in almost every peripheral tissue (i.e., the lungs, heart, liver, retina, and skeletal muscle) [8,9].

Circadian entrainment of the SCN by the daily photoperiod involves the transmission of a light signal from the photosensitive ganglionic cells in the retina mainly via the retinohypothalamic tract [10], which releases the excitatory neurotransmitter glutamate into the SCN. Other secondary afferents modulate this main excitatory signal, such as the gabergic neurotransmission conveyed by the geniculohypothalamic tract [11]. In response to these inputs, diverse efferent signals, both homeostatic and chronobiotic, are sent by the SCN to synchronize peripheral clocks. The former includes autonomic pathways sending information from the SCN to oscillators present in the adrenal gland and liver. This autonomic pathway modulates the sensitivity of the adrenal gland to the adrenocorticotropic hormone (ACTH) as well as the release of glucocorticoids [12,13], which act as humoral entraining signals for peripheral clocks [14,15]. The chronobiotic output sent by the SCN constitutes the main stimulus for the pineal gland’s circadian production rhythm of melatonin [16]. This hormone, whose concentrations peak at night, is rapidly released into the blood and cerebrospinal fluid, reaching all peripheral cells in the body [17].

At the molecular level, the core mechanism behind the cellular clockwork consists of a group of specific genes (‘clock genes’) and their products, which operate based on transcriptional/translational feedback loops (TTFLs) [18]. In mammals, the positive limb of this TTFL is represented by the heterodimeric partnership between two transcription factors: *CLOCK* (circadian locomotor output cycles kaput) and *BMAL1* (brain and muscle Arnt-like protein 1). During the day, the complex BMAL1:CLOCK translocates from the cytoplasm into the nucleus, where it activates E-box element-containing genes in the negative limb of the TTFL, promoting the expression of period (PER1, PER2, PER3) *and cryptochrome* (CRY1, CRY2) genes. After translation, PER and CRY proteins gradually start to accumulate in the cytoplasm and, with a delay of several hours, PER/CRY complexes translocate into the nucleus and inactivate the transcription of BMAL1:CLOCK dimer. This downregulates the expression of BMAL1:CLOCK [19]. The decrease in PER and CRY protein concentrations towards the end of the night phase leads to the disinhibition of BMAL1:CLOCK, reinitiating a new clock cycle [20,21]. Secondary TTFLs that stabilize the core TTFL have additionally been described. They include the nuclear receptors REV-ERB α/β (reverse erythroblastoma α/β; NR1D1/2) and ROR α/β (retinoid orphan receptor α/β), which act as transcriptional repressors and enhancers of BMAL1 expression, respectively [22,23].

### 1.2. Circadian Regulation of Physiological Functions: Immunity and Cognition

The efferent signals from the SCN and peripheral oscillators regulate the circadian succession of nearly all behavioral and physiological functions; additionally, core clock components participate in diverse intracellular pathways [9]. Apart from the sleep–wake cycle as the most prominent circadian rhythm [24], the circadian clock has been implicated in a variety of functions: cell cycle, apoptosis, and DNA damage and repair [25,26,27,28]; mammalian reproduction and the production of sexual hormones [29,30]; metabolism, as mutations in clock genes have been associated with weight gain, alterations in glucose, lipid metabolism, and insulin sensitivity [31,32,33]; mitochondrial function and redox status [34,35]; immunity and inflammation [36]; and brain function [37]. For the purpose of this article, we will be focusing on the last three functions.

#### 1.2.1. Circadian Regulation of Immunity and Inflammation

Several studies have pointed towards the link between immunity and the circadian clock. For instance, the mortality observed in mice exposed to constant darkness was three times higher than in those exposed to the standard light/dark cycle [38]. In humans, nocturnal and shift workers showed an increased susceptibility to infection [39,40]. Regarding innate immunity, the secretion of chromatin and antimicrobial molecules by neutrophils during NETosis (production of neutrophil extracellular traps), driven by the chemokine CXCL2, showed circadian variation both in mice and in humans [41]. In macrophages, knockout of BMAL1 gene was related to a higher survival rate in mice with streptococcal pneumonia [42]. BMAL1 has been shown to be a nuclear mediator of the innate immunity: on the one hand, it inhibited the expression of the chemokine (C-C motif) ligand 2 (Ccl2), reducing proinflammatory monocytes [43]; on the other hand, it induced the expression of RORα, which blocks the activation of the nuclear factor kappa β (NF-kβ) and, therefore, the production of proinflammatory molecules, such as interleukins 1-alpha (IL-1α) and 1-beta (IL-1β), tumor necrosis factor alpha (TNF-α), and prooxidant enzymes such as inducible nitric oxide synthase (iNOS) and cyclooxygenase-2 (COX-2) [44]. Contrarily, PER2 enhances the expression of INF-γ and IL-1β by blocking BMAL1:CLOCK activity [45]. Mitochondrial production of reactive oxygen species (ROS) changed throughout the day as well, according to animal [46] and human red blood cell studies [47].

The clock–immune interaction is also present in adaptive immunity. Genetic deletion of the BMAL1 gene blunted the rhythmic function of CD4+ T cells in mice [48]. Other components of the molecular clock, such as PER2 and REV-ERBα, showed oscillations in CD8+ T cells, as demonstrated through RNA sequencing [49]. The subset of T helper17 cells (Th17), which are involved in the defense against bacterial infections and the development of autoimmune diseases [50], changed in number according to the day/night cycle [51]. According to several studies in mice, REV-ERBα exerted negative feedback on Th17 cell development by influencing two transcription factors: RORγt, involved in Th17 differentiation [52]; and NFIL3, which downregulates RORγt [51]. The maturation of dendritic cells, which constitute the antigen-presenting cells [50], was reported to be negatively regulated by REV-ERBα/β as well [53].

The aforementioned findings suggest an involvement of the circadian clock in the pathophysiology of diverse chronic inflammatory processes. In a genetic association study on human patients, polymorphisms of PER3 were linked to a higher susceptibility to suffer both ulcerative colitis and Chron’s disease [54]; in a later study on dextran sulfate sodium-induced colitis in mice, those deficient in REV-ERBα were indeed more prone to suffer more severe symptoms based on the inactivation provoked by REV-ERBα on the NLRP3 inflammasome pathway [55]. Other examples of the link between the biological clock and chronic inflammatory diseases were provided in models of rheumatoid arthritis: patients suffering from this disease were reported to reach melatonin peak concentrations around 2 h earlier than healthy controls [56]; in murine models of induced arthritis, inflammatory markers significantly decreased during the dark phase, which was related to an inhibitory effect of CRY1 and 2 on inflammation within fibroblast-like synoviocytes [57]. Airway inflammation during asthma attacks also seems to be regulated by the molecular clock: in mice lacking BMAL1 in myeloid cells, induced attacks of allergic asthma were reported to be more severe than in wild-type controls and characterized by higher concentrations of chemokines (IL-4, IL-5) and eosinophils in the lung and serum; indeed, these results may explain the increase in asthma attacks at night, when the expression of BMAL1 is reduced [58]. Regarding neuroinflammatory processes, polymorphisms of BMAL1 and CLOCK were associated with higher risk for multiple sclerosis in a Slovenian study conducted on 900 patients and 1024 healthy controls [59]; the deletion of REV-ERBα favored pro-inflammatory cytokine expression mediated by Th17 cells, worsening experimental autoimmune encephalitis (EAE) and colitis [52].

#### 1.2.2. Circadian Regulation of Brain Function

The influence of circadian rhythms on behavioral and cognition was observed in studies conducted on patients with major depressive disorder and bipolar disorder, in whom symptoms seemed to improve after therapy with bright white light (>5000 lux, >30 min) during the day [60]. Contrarily, an association was reported between exposure to excessive light before bedtime and a higher risk for depressive symptoms [61]. In relation to animal models, exposure to light at night in rodents was linked to depressive-like symptoms without affecting circadian rhythms [62].

The interconnection between the biological clock and pathways implicated in memory consolidation was highlighted in studies conducted in nocturnal rodents. Their exposure to phase shifts was related to alterations in spatial learning and memory, suggesting an impact of circadian rhythm dysfunction (chronodisruption) on the hippocampal function [63,64]. In BMAL1-deficient mice, impairments in contextual fear and spatial memory were linked to deficits in the MAPK (mitogen-activated protein kinase)-signaling pathway [65]. In mice with a specific BMAL1 deficit in the hippocampus, disruptions in memory retrieval were related to dysregulation in dopamine D1 and D5 receptors [66]. Reductions in hippocampal neuronal plasticity were also observed in mice with PER2 mutations. Mutations in PER2 may be associated with a decreased activation of the transcription factor CREB (cAMP response-element binding protein) [67], involved in memory formation by reinforcing glutamatergic synapses [68].

Chronodisruption negatively impacts neurogenesis as well, as shown in adult mice suffering from chronic jet lag [69,70]. However, this effect was further reversed by the administration of melatonin [71], given its antioxidant properties [72] and the capacity of this hormone to induce the secretion of neurotrophic factors, such as insulin-like growth factor 1 (IGF-1) [73].

Based on the previously mentioned results, it is reasonable to consider the role of chronodisruption in the physiopathology of neurological diseases. CLOCK deletion in excitatory pyramidal neurons but not in inhibitory interneuron was followed by reduced seizure threshold and increased seizure frequency during sleep in mice [74]. BMAL1 deficiency in BMAL1 knock-out mice also contributed to epileptic excitability [75]. The effects of BMAL1:CLOCK on seizure frequency could be linked to their influence on the expression of the epilepsy-related transcription factors belonging to the PAR-domain basic leucine zipper (DBP, HLF and TEF) [74,75]. As a matter of fact, in murine models of N-methyl-D-aspartate (NMDA)-induced seizures, anticonvulsant effects of melatonin and adrenocorticotropic hormone (ACTH) were linked to significantly increased expressions of BMAL1, CLOCK, PER1, PER2, CRY1, and CRY2 [76]. With respect to investigations on murine models of Alzheimer’s disease (AD), Kress et al. [77] reported that the deposit of β-amyloid plaques in the hippocampus was exacerbated in BMAL1-deficient mice. In another study by the same group, BMAL1 deficiency was also associated with astrocyte activation and the trigger for inflammatory marker secretion [78], although the participation of astrocytes in plaque burden and, therefore, in the pathophysiology of AD remains controversial [79].

### 1.3. Dietary Fatty Acids and the Circadian Clock

#### 1.3.1. Omega-3 Fatty Acids and Their Effects on Cognition and Immunity/Inflammation

Omega-3 (ω-3) fatty acids are characterized by the presence of a double-bond between the third and fourth carbon atoms from the methyl end [80]. A representative subset is composed of ω-3 polyunsaturated fatty acids (ω-3 PUFAs), which contain double-bonds in *cis*-configuration [81]. ω-3 PUFAs consist of the essential fatty acid α-linolenic (ALA, 18:3), precursor of the long-chain fatty acids eicosapentaenoic (EPA, 20:5) and docosahexaenoic (DHA, 22:6) [82]. Fish constitute the primary source of EPA and DHA for humans [83].

ω-3 PUFAs are well-known for their effects on neurocognition and inflammation/redox status, functions that are, at the same time, under the control of the biological clock:Neurocognition: brain volume increase during the first two years of life occurs in association with the accumulation of ω-3 PUFAs in neuronal membranes [84]. Indeed, DHA constitutes the primary long-chain ω-3 PUFA in the brain, representing around 15% of total lipids [85]. ω-3 PUFAs modulate the fluidity of neuronal membranes [86] through their incorporation into the tails of phospholipids [87], and they participate in dendrite elongation, synaptic plasticity [88], and neurotransmission [89,90]. A ω-3 PUFA-enriched diet during pregnancy was reported to increase the expression of the brain-derived neurotrophic factor (BDNF), tropomyosin receptor kinase B (TRkB), and cAMP response-element binding protein (CREB), which benefit memory and behavior in offspring [91]. Lower serum DHA and EPA levels and higher ω-6: ω-3 ratios were reported in patients with autism spectrum disorder (ASD) and attention-deficit/hyperactivity disorder (ADHD) in comparison with typically developing controls [92,93]. In a recent study on two complementary mouse models of AD, intranasal administration of DHA reduced cognitive deterioration and Tau hyperphosphorylation (histopathological hallmark of AD) via inhibition of the c-Jun N-terminal kinase (JNK) [94].Inflammation/redox status: ω-3 PUFAs reduce platelet aggregation and are precursors of prostaglandins E3, which competitively block the conversion of the ω-6 arachidonic acid (AA) into highly inflammatory substances, such as prostaglandin E2 [95,96]. Diets rich in ω-3 PUFAs were observed to decrease the production of proinflammatory cytokines in peripheral blood mononuclear cells, such as IL-1, IL-6, and TNF-α, as well as to inhibit the expression of adhesion molecules, such as the monocyte chemoattractant protein-1 (MCP-1) [97,98,99]. Using cell culture media, Morin et al. [100] reported that the administration of ALA to psoriatic skin models was followed by reduced infiltration of T cells into the epidermis and decreased concentrations of the IL-6, IL-8, and the chemokine C-X-C motif ligand 1 (CXCL-1). DHA is used by macrophages to biosynthesize maresins, pro-resolving mediators with anti-inflammatory and anti-atherosclerotic actions [101]. One of these maresins, maresin-2 (MaR2), showed anti-inflammatory properties such as dexamethasone in murine models of asthma. These actions were apparently due to inhibition of the NLRP3 inflammasome pathway and the reduction in oxidative stress in lung tissue, with decreased levels of malondialdehyde (MDA) and increased concentrations of superoxide dismutase (SOD) and glutathione (GSH) [102]. In a randomized, double-blind, placebo-controlled trial on healthy volunteers, the administration of different doses of marine oil rich in ω-3 PUFAs was additionally associated with a reprogramming of the immune response, consisting of a dose-dependent increase in phagocytosis of bacteria by neutrophils and monocytes [103]. These anti-inflammatory/antioxidant properties have laid the foundation for using ω-3 PUFAs in patients recovering from surgery, proving to reduce postoperative complications (pyrexia, postsurgical infections, and wound dehiscence) and length of stay in hospital [104].

#### 1.3.2. Interactions between Dietary Fatty Acids and the Biological Clock

Diverse studies on animals and patients have pointed towards an influence of dietary fatty acids on circadian rhythms: taking into consideration that peroxisome proliferator-activated receptors (PPARs) are ligand-activated transcription factors involved in fatty acid metabolism and ketogenesis, Chikahisa et al. [105] assessed whether mice receiving chronic treatment with bezafibrate, a PPAR-agonist, experienced changes in sleep EEG recordings. An advance in the sleep/wake rhythm and an increase in slow-wave activity (SWA) during non-rapid eye movement sleep (NREM) were observed after bezafibrate administration. These changes occurred in parallel with variations in the serum concentrations of acetoacetate and β-hydroxybutyrate, ketone bodies peripherally used as an energy source when glucose is not readily available [106]. The reciprocal relationship between the sleep/wake cycle and fatty acid metabolism was suggested by the same group in a further study: considerably increased serum concentrations of ketone bodies and an increased ketone body ratio (acetoacetate/β-hydroxybutyrate) were observed in mice after sleep deprivation; at the same time, the intracerebroventricular injection of acetoacetate to mice significantly increased EEG SWA during NREM sleep [107]. Regarding clinical studies, Siegmann et al. [108] performed a non-randomized longitudinal trial to compare sleep-quality measures between type-2 diabetes and prediabetes patients following a continuous care intervention focused on nutritional ketosis and those patients with type 2 diabetes receiving the usual standard of care. Global sleep quality, as evaluated through the Pittsburgh Sleep Quality Index (PSQI), significantly improved in the group enrolled in the continuous care intervention compared with the patients receiving the usual care, suggesting a beneficial effect of nutritional ketosis on sleep.

Tognini et al. [109] revealed that the effects of a ketogenic diet on murine peripheral clocks were tissue-dependent. A ketogenic diet is a high-fat, adequate protein, very-low-carbohydrate diet that induces a switch to fatty acid oxidation as an energy source. Ketogenic diet-fed mice showed increased PPARα in both liver and gut tissues, but with a marked circadian variation (diurnal) only in the latter; these results were paralleled by circadian oscillations of serum and intestinal β-hydroxybutyrate concentrations. The biological clock modulates the metabolic and inflammatory actions derived from dietary fatty acids as well: variations in serum triglyceride concentrations after the administration of different fats/oils (butter, olive oil, and fish oil) to male Wistar rats changed according to the circadian time (zeitgeber time or ZT) at which they were given [110]. In a recent study performed on the Korean population to determine risk for obesity based on macronutrient intake patterns, Shon et al. [111] found that a very low fat-to-carbohydrate ratio (FC) was linked to a higher risk for obesity than the optimal FC, and this association was in turn modulated by specific polymorphisms of CLOCK, PER2, and CRY1.

### 1.4. Justification and Aim

According to the previous findings, it is reasonable to consider that an interaction must exist between the molecular circadian clock and ω-3 fatty acids, as one of the most bioactive fatty acids in mammals [112]. On the grounds of the effects of ω-3 fatty acids on the inflammatory process and cognition, which are functions also controlled by the molecular clock, ω-3 fatty acids could act as circadian rhythm synchronizers. The current work aims at providing a comprehensive review of the existing literature on the role of ω-3 fatty acids in circadian rhythm regulation and its molecular mechanisms.

### 1.5. Search Strategy

Although not intended to conduct a systematic review, our literature search was performed on PubMed, Scopus, Ebsco, and OvidSP following pre-defined selection criteria. Our objective was to identify articles providing original evidence on the regulating actions exerted by ω-3 fatty acids on the molecular circadian clock, published between January 1970 and April 2022. The search terms (MeSH) for the molecular clock encompassed ‘chronobiology disorders’ OR ‘circadian clocks’ OR ‘circadian rhythm signaling peptides and proteins’ OR ‘CLOCK proteins’ OR ‘period circadian proteins’ OR ‘biological clock’ OR ‘circadian rhythm’. The search terms (MeSH) for ω-3 fatty acids included ‘fatty acids, omega-3′ OR ‘fatty acids, n-3′. Inclusion criteria consisted of: (1) studies focused on the effect of ω-3 fatty acids on circadian rhythms; (2) observational prospective and retrospective studies, cross-sectional studies, experimental studies, systematic reviews, and meta-analyses; (3) pre-clinical studies (conducted on both animals and cell cultures) and clinical trials (in humans); and (4) articles written in English.

A total of 3500 articles were identified through a database search. Based on the title and abstract, 3477 manuscripts were removed, as they were focused on fatty acids different from ω-3 fatty acids, did not explore interactions between ω-3 fatty acids and the circadian clock, or the full text was not written in English. Twenty-three articles were assessed: after rejecting two articles whose full text was not found, 21 articles were finally included in this review.

## 2. Regulation of Melatonin Secretion by Omega-3 Fatty Acids

Gazzah et al. [113] investigated differences in adenosine-dependent melatonin release in cultured pineal glands from 2-month-old rats fed with either peanut oil (ω-3-deficient diet) or peanut plus rapeseed oil (ω-3 adequate diet). A significant decrease in total pineal DHA and the proportion of DHA in the pineal phospholipids phosphatidylcholine and phosphatidylethanolamine were found in the ω-3-deficient group, accompanied by a significantly lower release of melatonin.

Zhang et al. [114] assessed the lipid profiles, phospholipid molecular species, lipoxygenase metabolites, and melatonin levels in the pineal glands of 12-month-old offspring of Sprague–Dawley rats fed with either an ω-3-deficient or an ω-3-adequate diet. In the ω-3-deficient group, the authors reported a significant decrease in pineal DHA concentrations, accompanied by a significant increase in pineal total levels of ω-6 fatty acids, particularly AA. Similar to Gazzah et al. [113], a significant reduction in DHA was also observed in the composition of pineal phospholipids from the classes phosphatidylcholine, phosphatidylethanolamine, and phosphatidylserine in the offspring of rats receiving the ω-3-deficient diet. At the same time, and in contrast to Gazzah et al. [113], a negative association was found between DHA concentrations and melatonin levels in the pineal gland; indeed, a significant increase in diurnal pineal melatonin levels occurred in parallel with the reduction in pineal DHA quantities in the ω-3-deficient group. The authors attributed the increase observed in the pineal melatonin concentrations to alterations in receptors and enzymes involved in melatonin synthesis secondary to changes in the ω-6/ω-3 ratio; alternatively, reductions in lipoxygenase metabolites, such as 12-hydroxy eicosatetraenoic acid (12-HETE), in ω-3-deficient fed rats, could have disrupted the melatonin synthesis pathway. The study design did not allow for assessing changes in the circadian variation of pineal melatonin concentrations, as the authors only performed one daily measurement of melatonin (during the light phase).

In a later investigation, Zaouali-Ajna et al. [115], the researchers from Gazzah’s team, performed 12 h measurements (daytime and nighttime) of urinary excretion of 6-sulfatoxymelatonin (aMT6), the main metabolite of melatonin, in the 2-month-old descendants of rats receiving either a control or an ω-3-deficient diet through a second-generation model. The authors found significantly lower nighttime urinary aMT6 concentrations in the ω-3-deficient group compared with the control group; however, no significant differences were found in the daytime concentrations. The administration of DHA-enriched phospholipids for 5 weeks seemed to restore melatonin production in the ω-3-deficient group, as observed by the restoration of nocturnal urinary aMT6 excretion. Considering that urinary aMT6 concentrations reflect plasma melatonin levels and, therefore pineal melatonin production, since this hormone is immediately released from the pineal gland after being produced, these results corroborated the authors’ previous studies [113] and differed from Zhang et al. [114].

The findings obtained by Zaouali-Ajna et al. [115] were further supported by Lavialle et al. [116], who compared the pineal melatonin rhythm, locomotor activity, and dopaminergic neurotransmission in the striatum of 3-month-old male Syrian hamsters that received an ω-3-deficient diet with those that received an ω-3-adequate diet. In the ω-3-deficient group, a doubling of the AA/DHA ratio in the striatum and an even greater increase in the pineal gland was accompanied by a significant reduction in the nocturnal melatonin secretion peak. Nevertheless, no significant differences in pineal melatonin concentrations were observed between groups during the light phase. The wheel-running test evidenced a significantly higher locomotor activity in the hamsters receiving the ω-3-deficient diet, which occurred in parallel with a significant increase in the concentrations of dopamine and dopaminergic metabolites (homovalinic acid and 3,4-dihydroxyphenylacetic acid) in the striatum. As proposed earlier by Zhang et al. [114], the authors indicated that the decrease in nocturnal pineal melatonin concentrations may be related to a reduction in 12-HETE levels, as fewer 12-HETE concentrations are associated with diminished non-esterified AA and, therefore, higher AA quantities in the pineal gland. As per the hyperlocomotion found in the ω-3-deficient hamsters, the researchers pointed to a combination of factors contributing to striatal hyperdopaminergic: on the one hand, reduced pineal melatonin concentrations, as melatonin seems to inhibit the release of dopamine in the brain [117]; on the other hand, membrane-bound proteins, such as enzymes, receptors, and transporters, including dopamine transporters, could have been inactivated as a result of diminished DHA levels, given that DHA is the main component of neuronal membranes.

Table 1 summarizes the aforementioned articles according to study design, population, experimental diets, comparison, and outcome.

## 3. Regulation of Inflammatory Pathways by Omega-3 Fatty Acids via Clock Genes

In a murine neuronal model study, the hypothalamic cell line mHypoE-37, with a robust cellular circadian machinery, was used by Greco et al. [118] to investigate the changes induced by dietary fatty acids on the molecular circadian clock and possible interactions with inflammatory pathways. The cells were divided into groups that received one of four treatments: (1) no treatment; (2) treatment with the saturated fatty acid palmitate; (3) treatment with DHA; and (4) pre-treatment with DHA followed by combined treatment with palmitate plus DHA. After treatment, the following parameters were assessed: expression and circadian profiles of the clock genes BMAL1, PER2, REV-ERBα, and CRY; expression of the proinflammatory cytokine IL-6 and the toll-like receptor 4 (TLR4), which represents the canonical pathway to activate the NF-kβ; and the Agouti-related protein (AgRP), an orexigenic neuropeptide involved in energy homeostasis. A significant upregulation of BMAL1 transcript levels was observed following treatment with palmitate in comparison with the control group, together with a significant upregulation of REV-ERBα. Palmitate induced a non-significant phase delay in BMAL1 and REV-ERBα, with no significant changes in their period or amplitude. No significant differences were reported in the circadian profiles (period, amplitude, and acrophase of the cycle) of clock gene expression between the palmitate and the control groups. Further, no significant changes in the transcript levels of IL-6, TLR4, and AgRP were observed after treatment with palmitate, which seemed to indicate that alterations of the hypothalamic molecular clock occur through mechanisms different from those that operate during hypothalamic cellular inflammation. Cells treated with DHA showed a significant upregulation of BMAL1, accompanied by an extension and phase-advance of the BMAL1 period. No comments about the changes induced by DHA on the other clock genes were made by the authors. Regarding inflammatory genes, a significant reduction in the expression of the cytokine IL-6 was observed in cells treated with DHA when compared with the other groups. The significant increase provoked by palmitate in BMAL1 expression was reduced when the cells were pre-treated with DHA, which suggested that DHA may have a protective effect against the changes induced by saturated fatty acids on the molecular clock. Additionally, the authors assessed changes in the phosphorylation of glycogen synthase kinase 3β (GSK3β) and adenosine monophosphate-activated protein kinase (AMPK). GSK3β enhances the expression of BMAL1 by altering the stability of REV-ERBα [119], whereas AMPK is an NF-kβ repressor and an inhibitor of both PPARα and PPARγ, therefore reducing the expression of BMAL1 [120]. No changes in the phosphorylation status of either GSK3β or AMPK were observed after treatment with palmitate or DHA. Accordingly, the authors concluded that hypothalamic circadian clock disruption was independent of GSK3β/AMPK. They finally stated that BMAL1 upregulation by dietary fatty acids could be PPAR-dependent, although the exact mechanisms through which saturated fatty acids and ω-3 PUFAs have differential effects on the circadian clock are yet to be elucidated.

In line with Greco et al. [118], Kim et al. [121] assessed the clock oscillatory profile variations induced by palmitate and DHA in two peripheral cells, *Bmal1-dLuc* fibroblasts and adipocytes (differentiated from fibroblasts), as well as timekeeping associations with changes in IL-6 and NF-kβ expression. Acute administration of DHA at different time points induced small phase delays in both *Bmal1-dLuc* fibroblast and adipocyte rhythms in comparison with the control medium (bovine serum albumin), although the maximal phase shifts were reached at different time points in each cell type. In good agreement with Greco et al. [118], DHA treatment was associated with a significant reduction in IL-6 expression but did not affect NF-kβ activation. The anti-inflammatory effects of DHA were, indeed, maximal at the cell-dependent time points at which phase shifts were most evident. Acute treatment with palmitate produced the opposite: large phase advances that were also maximal at different time points for each cell type. Regarding the inflammatory response, palmitate induced significant increases in NF-kβ activation and IL-6 expression, which coincided with the maximal phase shifts for each cell type. Additionally, the authors evaluated the effects of different anti-inflammatory stimuli, such as DHA and AICAR, an AMPK activator, to reduce the phase-shifting actions of palmitate. Both DHA and AICAR attenuated the inflammatory cascade as well as the circadian phase alterations provoked by palmitate. This study disproves Greco et al.’s results, as it points to time-dependent interconnections between molecular clock alterations and inflammatory signaling, as well as to a possible intervention of AMPK phosphorylation in the chronodisruption induced by palmitate.

Using an RNA sequencing-based transcriptomic approach, Yuan et al. [122] investigated potential anti-inflammatory molecular mechanisms underpinning the metabolic protective effects of ω-3 fatty acids in a Western-style diet-induced non-alcoholic fatty liver disease (NAFLD) rat model. NAFLD, the most common chronic liver disease in developed countries, is believed to be induced by the Western-style diet, in which high contents of cholesterol and saturated-free fatty acids lead to steatohepatitis through lipoapoptosis and an inflammatory cascade involving the TLR4 [123]. Eight- to nine-month-old male Sprague–Dawley rats were randomly assigned to three different dietary groups for 16 weeks: (1) the first group received a standard chow diet (control); (2) the second group received a Western-style, high-fat, high-cholesterol, and high-sugar diet (WD); and (3) the third group received a Western-like diet but containing ω-3-enriched fish oil (FOH). Serum lipid and glucose profiles and hepatic transaminases were measured after treatment; a liver histological examination to evaluate steatosis was obtained, as well as the expression of clock genes and genes involved in lipid metabolism and inflammatory immune response. Triglycerides, total cholesterol, low-density lipoprotein cholesterol (LDL-C), and non-esterified fatty acid concentration in plasma were significantly higher in the WD in comparison with controls. A significant increase in plasma levels of the transaminases alanine aminotransferase (ALT) and aspartate aminotransferase (AST) was also found in the WD group in relation to controls. All these serum parameters were significantly reduced by fish oil feeding. The protective effects induced by the fish oil diet were also reflected histologically: WD feeding induced severe macrovesicular steatosis and inflammatory cell infiltration, which were significantly alleviated in the biopsies belonging to the FOH group. The significantly higher hepatic levels of IL-1β and TNF-α found in WD-fed rats were also reversed by fish oil. On a molecular level, WD significantly increased the expression of the sterol regulatory element binding factor 1 (Srebf1), suppressor of cytokine signaling 2 (Socs2), chemokine Ccl-2, and semaphorin 4A (Sema4a). Srebf1 is involved in de novo lipogenesis, whereas Socs2 is a key molecule in the JAK-STAT-SOC pathway, implicated in the transcription of pro-inflammatory cytokines and tumorigenesis [124]; Ccl-2 favors hepatic steatosis and inflammation [125], and Sema4a participates in the differentiation of T-helper lymphocytes during hepatic inflammatory cell infiltration [126]. The increased expression levels of these genes were significantly attenuated by fish oil feeding. The inflammatory and lipid metabolism changes in the WD group were accompanied by a significant raise in the expression of BMAL1, together with a significant downregulation of PER2 and PER3 expression in comparison with controls. Clock disturbances were reversed in the FOH. Taken together, Yuan et al.’s results suggest that the protective effects of ω-3 fatty acids on hepatic lipid metabolism are mediated via the regulation of clock gene expression.

Following the NAFLD model, Han et al. [127] added evidence to the protective effects of ω-3 fatty acids by exploring the properties of maresin-1 (MaR1) to induce the anti-inflammatory phenotype (M2 polarity) of liver macrophages via RORα binding. NAFD was induced through a high-fat diet in both wild-type and myeloid-specific RORα knockout mice. After treatment with DHA, an increased expression of RORα-dependent M2 polarity shift was observed in liver macrophages. These changes did not occur in DHA-treated myeloid-specific RORα knockout mice. The authors reported that the transcriptional activation of RORα by DHA was mediated by MaR1, a DHA metabolite produced by macrophages through different enzymes, such as 12-lipoxygenase (12-LOX) [128]. Treatment with MaR1 induced the expression of RORα by favoring its binding to a coactivator p300 and, secondarily, modifying active histones. The increased transcription of RORα was accompanied by a switch to M2 polarity in liver macrophages that did not take place in MaR1-treated RORα knockout mice, proving that the M2 polarity shift in liver macrophages was RORα-dependent. However, other activities induced by MaR1 in liver macrophages, such as phagocytosis, did not show RORα-dependence, as MaR1-induced phagocytosis also occurred in the RORα knockout group. MaR1-provoked actions led to decreased liver injury markers and attenuated fibrotic changes (collagen deposit, lipid peroxidation, expression of profibrotic proteins) in the liver that were not observed in the RORα knockout mice that were treated with MaR1. It is interesting to note that the actions induced by MaR1 seemed to be relatively specific to RORα, as the transcriptional activity of other nuclear receptors, such as RORβ, RORγ, PPAR, retinoid X receptor α, and liver X receptor α experienced no variations with MaR1 treatment. Additionally, the authors reported that treatment with SR1078 (a synthetic RORα agonist), MaR1, or adeno-associated virus RORα increased the level of 12-LOX-encoding platelet type 12-LOX (Alox12) mRNA, whereas knockdown of RORα decreased these levels. These results suggest that RORα may influence MaR1 biosynthesis via Alox 12; thereby, MaR1-RORα-12-LOX would constitute an autoregulatory circuit loop supporting the protective effects of ω-3 fatty acids.

The previously mentioned studies are represented in Table 2.

## 4. Regulation of Neurological Functions by Omega-3 Fatty Acids via Clock Genes

Based on previous works showing that stroke-prone spontaneously hypertensive rats (SHRSP) tended to show behavioral abnormalities at the onset of stroke, including increased ambulatory activity and disrupted circadian rhythms [129], Minami et al. [130] assessed the effects of DHA administration on the stroke-related behavioral changes and lifespan in SHRSP. SHRSP were randomly assigned to three types of diets during their lifespan: (1) a control diet without DHA; (2) a diet containing 1% DHA, and (3) a diet containing 5% DHA. DHA was found to significantly suppress the gradual increase in systolic blood pressure in SHRSP in a dose-dependent manner. This led to a prolonged survival time in DHA-treated SHRSP and a drift in the most prevalent death cause from cerebral infarction in control diet-fed SHRSP to senility in the DHA-fed group. The ambulatory activity was measured during continuous 3 h intervals, observing a circadian desynchronization with the dark and light phases in control diet-fed SHRSP, with a significant increase in ambulatory activity in the light vs. dark phase. However, the ambulatory activity in DHA-treated SHRSP showed a preserved circadian pattern, with significantly higher activity in the dark vs. light phase, as it corresponds to nocturnal animals. In further brain tissue analyses, significantly reduced plasma, and cortex AA levels together with significantly higher plasma and cortex DHA concentrations were found in DHA-fed SHRSP compared with the control-fed group.

Umezawa et al. [131] compared differences in learning ability and natural behavior in 15-month-old senescence-resistant SAMR1 mice, a murine model with no age-related deterioration of memory and learning, after being fed with either an α-linolenate-rich diet (ω-3-enriched diet) or an α-linolenate-deficient diet (ω-3-deficient diet). Perilla oil, which contains an ω-6/ω-3 ratio of 0.24, was used for the ω-3-enriched diet; safflower oil, with an ω-6/ω-3 ratio of 250.3, was chosen as the ω-3-deficient diet. In line with Lavialle et al. [116], the amount of locomotor activity recorded over 5 min in the open-field test was found to be significantly higher in the safflower group. Observation of the circadian rhythm of spontaneous motor activity in 48 h revealed a significant increase in activity in the first and second dark periods, as well as in the second light period, in the safflower oil-fed mice in relation to the perilla oil group. Nevertheless, the circadian pattern (light vs. dark phase) of spontaneous motor activity was preserved in both groups, which is in contrast to Minami et al. [130]. In the light and dark discrimination tests, by which positively and negatively rewarded stimuli were repeatedly presented to mice after pressing a lever, the number of incorrect responses markedly decreased as the session continued in the perilla oil-fed mice in comparison with those fed with safflower oil. Later assessment of brain lipids revealed that the differences in ω-6/ω-3 ratios between the diets were reflected in the (ω-6/ω-3) fatty acid ratios in the brain phospholipids phosphatidylethanolamine and phosphatidylcholine: the proportions of total ω-3 PUFAs, including DHA and EPA, in brain phosphatidylethanolamine and phosphatidylcholine were significantly higher in the perilla oil group than in the safflower oil group. These results highlight that the dietary ω-6/ω-3 fatty acid ratio modifies the composition of brain phospholipids, having a subsequent impact on the circadian pattern of locomotor activity and discriminative learning ability.

Fiala et al. [132] performed a clinical trial to investigate the effects of ω-3 fatty acid supplementation on cognition, immunity, transcription of energy, and clock gene expression in a cohort of neurodegenerative patients who were refractory to treatment with cholinesterase inhibitors (CI) or other standard therapies. The cohort included 28 patients diagnosed with subjective cognitive impairment (SCI) (*n* = 8), mild cognitive impairment (MCI) (*n* = 8), Alzheimer’s disease (AD) (*n* = 6), frontotemporal dementia (FTD) (*n* = 1), vascular cognitive impairment (VCI) (*n* = 2), and dementia with Lewy bodies (DLB) (*n* = 3). Nutritional supplementation included the ω-3 drink Smartfish, containing DHA (5 mg/mL), EPA (5 mg/mL), and botanical antioxidants. A significant cognitive improvement according to the Mini-Mental State Examination (MMSE) scores was found in MCI patients when receiving an add-on ω-3 supplement to CI therapy, which subsequently delayed progression to dementia by 26 to 54 months. The immune function of peripheral blood mononuclear cells (PBMC) was assessed by examining amyloid-β phagocytosis by macrophages in blood. A significant increase in amyloid-β phagocytosis was observed in MCI patients after supplementation, which occurred in parallel with upregulation of energy genes in PBMC, such as enzymes involved in glycolysis (hexokinase and glyceraldehyde 3-P dehydrogenase or GAPDH), tricarboxylic cycle (α-ketoglutarate or OGDH), and nicotinamide adenine dinucleotide (NAD) metabolism. These results were accompanied by changes in clock gene expression in PBMC of MCI patients after ω-3 supplementation: in a trans-sectional analysis in vitro comparing placebo vs. ω-3 on the same visit, an upregulation of the circadian regulator aryl hydrocarbon receptor nuclear translocator-like 2 (ARNTL2), which suppresses the transcription of the proinflammatory IL-21 gene [133], was observed; longitudinal analysis on two successive visits (ω-3 on visit 1 and visit 2) revealed a non-significant upregulation of CLOCK by ω-3 fatty acids. Interestingly, a 5-year follow-up of one of the MCI patients showed gradual cognitive decline paralleled with a flat trend in CLOCK/ARNTL2 concentrations following the initial cognitive improvement in the first 26 months.

Table 3 lists the articles found on the neurological effects derived from circadian regulation by ω-3 fatty acids.

## 5. Other Studies Supporting a Regulatory Effect of Omega-3 Fatty Acids on Circadian Rhythms

### 5.1. Regulation of Metabolic Functions by Omega-3 Fatty Acids via Clock Genes

Because microRNAs (miRNAs) have been reported to have a substantial metabolic role in hepatic metabolism [134], cardiovascular function [135], and insulin signaling [136], Daimiel-Ruiz et al. [137] conducted an in vitro epigenetic-approached study to investigate the effects of different dietary lipids on human enterocytes miRNA profile. Differentiated Caco-2 cells were used and treated with micelles containing cholesterol, DHA, or conjugated linoleic acid. Microarray assessment revealed treatment-induced modifications in the expression levels of 29 miRNAs; among them, miR-107 was upregulated by all three treatments, but the increase was only statistically significant after DHA treatment. A later in silico analysis of a 2 Kb transcription region in mi-R107 predicted the presence of binding sites for transcription factors with involvement in lipid metabolism, such as myogenin, c-Fos/c-Jun, PPARα, and sterol regulatory element-binding protein (SREBP). The inhibition of miR-107 induced a significant increase in the expression of CLOCK, accompanied by a significant decrease in the expression of CRY2. An overexpression of miR-107 also increased CLOCK levels, as well as PER2 levels, whereas it reduced CRY2 levels. It is of interest that both inhibition and overexpression of mi-R107 increased CLOCK mRNA levels. However, when analyzing the effects of miR-107 on CLOCK protein levels, CLOCK protein concentrations were only upregulated after mi-R107 inhibition. On the contrary, miR-107 overexpression decreased CLOCK protein levels in a time-dependent manner. Given these results, the authors proposed that miR-107 may jeopardize CLOCK mRNA translation; therefore, increased CLOCK mRNA levels after mi-R107 overexpression could be the result of cellular response to compensate for diminished concentrations of CLOCK protein. Additionally, the luciferase activity assay revealed that miR-107 directly binds to CLOCK. In further analyses, the authors reported that the transfection of Caco-2 cells with syn-miRNA-107 (miR-107 mimic) induced modifications in the circadian rhythms of CLOCK and CLOCK-related genes (PER and CRY), including changes in the acrophase and amplitude. Together, these results highlight the miRNAs as an additional mechanism to explain the metabolic benefits derived from ω-3 fatty acids and add evidence to the role played by clock genes and circadian rhythms in the interaction between ω-3 fatty acids and metabolic pathways.

Furutani et al. [138] aimed to assess the ability of fish oil or a DHA/EPA-containing diet to induce phase shifts of the liver clock in C57BL/6J mice undergoing restricted feeding, as well as to evaluate the role played by insulin secretion in this interaction. Restricted feeding with various fish oil-containing diets (sardine, menhaden, saury, and tuna) induced a significant liver clock phase delay in comparison with restrictively fed mice with a control chow diet. Liver clock phase delay was larger in the tuna group than in the groups fed with other types of fish oil. Considering that this difference could have been due to the higher proportions of DHA/EPA contained in tuna oil vs. the other fish oils, the authors further tested the magnitude of liver clock phase shifts after substituting the soybean oil component of the chow diet with DHA/EPA in similar proportions to the ones found in tuna oil: a large phase delay (around 4.3 h) of the liver clock was observed, similar to the one observed with tuna oil. The liver clock phase shifts induced by either tuna oil or DHA/EPA-supplemented chow diet occurred in parallel with a significant increase in blood insulin levels and PER2 expression in the liver. The essential role played by insulin release in the phase-shifting effects of tuna oil was further corroborated in streptozotocin-induced insulin-depleted mice, in which the liver clock phase experienced no change after administering tuna oil. Additionally, the researchers examined the participation of GPR120 (FFAR4), a long-chain fatty acid receptor highly expressed in the adipose tissue and activated by ω-3 fatty acids, which are involved in anti-inflammatory pathways and insulin sensitivity [139]. Tuna oil phase delay, insulin secretion, and induction of PER2 expression were attenuated in GPR120-deficient mice. Finally, the circadian effects derived from administering tuna oil or DHA/EPA-containing chow diet were explored under free-feeding conditions, revealing solely a weak effect on circadian phase shift and blood insulin concentrations. These findings suggest that fish oil and DHA/EPA may be key elements in the restricted feeding-induced entrainment of the liver clock by favoring insulin release via GPR120.

Amos et al. [140] compared the effects on redox status and metabolic homeostasis of a high-fat diet (HFD) and an ω-3-enriched diet (OM3) in the ‘stress-less’ Bob-Cat mice, a murine model overexpressing the antioxidant catalase [141]. Despite sex-related differences, eight weeks of dietary intervention showed that Bob-Cat mice fed with OM3, in contrast to HFD, maintained or lowered body weight and fat mass. The OM3-diet also stabilized glucose levels and insulin homeostasis, as measured by the Calculated Homeostasis Model Assessment for Insulin Resistance (HOMA-IR) and levels of adiponectin, a hormone produced by the adipose tissue that alleviates insulin resistance [142]. These changes occurred in parallel with a significant increase in the expression of GPR120/FFAR4. A more satisfactory regulation of redox status was revealed in the OM3 group compared with the HFD-fed mice, as evidenced by significantly higher levels of mRNA expression of the nuclear factor E2-related factor 2 (Nrf2), a transcription factor that activates both antioxidant catalase and heme oxygenase (HO-1) [143]. Although no effects on specific clock genes were assessed, based on the pattern of food intake (light vs. dark cycle), the researchers reported preserved circadian rhythms in OM3-fed mice in contrast to the HFD group. These results corroborate previous findings by Furutani et al. [138] by providing additional evidence on the implication of GPR120/FFAR4 and Nrf2 in the interaction between ω-3 and circadian rhythms.

### 5.2. Regulation of Reproductive Functions by Omega-3 Fatty Acids via Clock Genes

We found one study by Wang et al. [144] evaluating the beneficial effects derived from ω-3 fatty acids to alleviate high-fat diet-induced reproductive dysfunction and restore circadian gene expression in male mice. Seven-week-old male C57BL/6J mice were randomly allocated to three different diets for 12 weeks: (1) a standard chow diet (control), with 53% ω-6 fatty acids and 1% ω-3 fatty acids; (2) a high-fat diet (HFD), with 17.14% ω-6 fatty acids and 0.33% ω-3 fatty acids; and (3) a fish oil-enriched diet (FO), with 10.98% ω-6 fatty acids and 10.96% ω-3 fatty acids. Body weight, perinephric fat weight, and perinephric and epididymal fat weight/body weight ratios were significantly higher in both the HFD and in the FO groups compared with the control group. Plasma total cholesterol and triacylglycerol concentrations were found to be significantly increased in HFD-fed mice vs. controls and significantly decreased in FO-fed mice compared with the HFD group. Histological examination of testes in the HFD group revealed the increased size of seminiferous tubules, altered arrangement of spermatogenic cells, reduced sperm, significantly decreased Leydig cells, and a higher apoptosis index in comparison with controls. Accordingly, the mRNA levels of proteins involved in testosterone synthesis were significantly downregulated in the HFD group, such as the luteinizing hormone receptor (LHR); the steroidogenic acute regulatory receptor (STAR), which transports cholesterol to the inner mitochondria membrane; the cholesterol side-chain cleavage enzyme (P450scc), which represents the rate-limiting enzyme in testosterone synthesis by catalyzing the oxidative cleavage of the side-chain of cholesterol to form pregnenolone [145,146,147]; 3β-hydroxysteroid dehydrogenase (3β-HSD), which transports pregnenolone to the smooth endoplasmic reticulum; 17β-hydroxysteroid dehydrogenase (17β-HSD), which reduces pregnenolone into testosterone [147,148]; and the steroidogenic factor 1 (SF-1), a member of the orphan nuclear receptor family that stimulates the expression of STAR and all other genes involved in the production of testicular steroids [149,150]. The histological modifications and testosterone synthesis disruption observed in HFD-fed mice were reversed by the FO diet, as shown by restored expression levels of the aforementioned genes (except for 17β-HSD concentrations) and an increased plasma testosterone/estradiol ratio in FO-fed mice. To evaluate circadian oscillations of testosterone synthesis and testicular clock genes, the authors assessed the mRNA expression levels of LHR, STAR, P450scc, 3β-HSD, 17β-HSD, SF-1, BMAL, CLOCK, CRY1, CRY2, and PER2 at zeitgeber time 0 (ZT0, in the morning) and zeitgeber time 12 (ZT12, at night): no remarkable differences in the expression levels of these genes were observed between ZT0 and ZT12 in the control group; however, these differences were found to be significant in the HFD group, together with a significant generalized downregulation in the expression of BMAL1, CLOCK, PER2, and CRY2; the alterations found with HFD were alleviated by FO.

### 5.3. Circadian Regulation of Cardiovascular Functions by Omega-3 Fatty Acids

Considering heart-rate variability (HRV) as a predictor of sudden death in patients with coronary heart disease (CHD), Mozzafarian et al. [151] conducted a cross-sectional study in a population-based cohort of 4465 men and women ≥65 years of age to investigate associations between usual dietary consumption of ω-3 fatty acids and measures of HRV. Dietary intake was assessed through a food-frequency questionnaire referred to the previous year. HRV measures were evaluated through resting 12-lead electrocardiograms (ECG) and 24-h Holter monitor recordings: parasympathetic (respiratory) variations, circadian fluctuations, and erratic HRV due to abnormal firing of the sinoatrial node were tested. After adjustment for demographic variables, clinical risk factors, lifestyle, and dietary characteristics, the authors reported significant associations between ω-3 fatty acid intake and measures reflecting vagally mediated respiratory variation. At the same time, inverse associations were found with measures reflecting abnormal heart rate patterns due to erratic firing of the sinoatrial node. The researchers further carried out a 10-year follow-up of the sample to explore associations between HRV parameters and risk of CHD death, revealing a significantly lower relative risk of CHD death with increased vagally mediated HRV and reduced abnormal sinotrial firing. Interestingly, no significant associations were found between dietary consumption of ω-3 fatty acids and measures of HRV circadian variation.

Buchhorn et al. [152] explored the effects of ω-3 fatty acid supplementation on 24 h Holter ECG-derived heart rate and HRV measures in a clinical sample of 18 children and adolescents diagnosed with attention-deficit disorder with and without hyperactivity (AD(H)D). Omega-3 fatty acid supplementation for 143.7 days on average had a significant inverse association with heart rate and a positive association with HRV in time-domain (rMSSD) and frequency-domain measures (high-frequency HRV via fast Fourier transformation over the frequency of 0.15–0.4 Hz). Significant interactions of ω-3 fatty acids and the time of day were found in the mean heart rate, rMSSD, and HRV. The circadian variation pattern of HRV was also assessed through cosine function parameters such as MESOR (the rhythm adjusted 24-h mean level), amplitude (the distance between MESOR and the highest maximum value of the cosine curve), and acrophase (the phase shift of amplitude from a given reference time point when the highest oscillation is reached). The results slightly differed from Mozzafarian et al. [151], as ω-3 fatty acid supplementation was found to have a significant effect on several of the parameters evaluating circadian variation pattern of HRV: MESOR of mean HR, rMSSD, and HF, and amplitude of mean HR and rMSSD. However, no effect was observed on the amplitude of HF and acrophase of any of the aforementioned measures. Although this study allowed for exploring temporal relationships between ω-3 fatty acid supplementation and changes in heart rate and HRV, it also had considerable limitations, such as no reports on combined psychostimulant treatment or compliance with ω-3 fatty acid supplements, as well as a lack of information on the brands or EPA/DHA content of the supplements used.

In a double-blind controlled trial conducted on 152 patients diagnosed with CHD and essential hypertension (HT), Bhardwaj et al. [153] aimed at exploring the effects of blended flaxseed oil (260 mg/mL of ω-3 fatty acids; 130 mg/mL of ω-6 fatty acids) on 24 h, 7 day circadian variations in blood pressure. Groups 1 (n = 50) and 2 (n = 50), respectively composed of patients with CHD and HT, received blended flaxseed oil for 24 weeks; group 3, composed of patients with CHD/HT, received sunflower oil (ω-3 fatty acids: 0.2 mg/mL; ω-6 fatty acids: 39.8 mg/mL) for the same period. Twenty-four-hour, seven-day continuous ambulatory blood pressure monitoring was carried out initially and after six months of supplementation. Results were processed through cosinor analysis, determining the mean value of the sinusoid (MESOR), the amplitude of circadian fluctuation, and acrophase (highest values in 24-h day). MESOR of both systolic (SBP) and diastolic blood pressure (DBP) significantly decreased in groups 1 and 2 after 6 months, together with a significant reduction in the MESOR of mean HR, as observed in the previous study by Buchhorn et al. [151]. The circadian amplitude and night–day differences in SBP/DBP were significantly reduced in groups 1 and 2 after supplementation, whereas no change was observed in group 3. The acrophase significantly changed at 6 months in group 2, although no significant differences in the acrophase were observed in the other groups. Together, these results suggest a protective role of ω-3 fatty acids in the synchronization of disrupted circadian patterns of blood pressure and autonomic regulation of the cardiovascular system in subjects with CHD and HT.

### 5.4. Circadian Regulation of Biochemical Processes by Omega-3 Fatty Acids

Mollard et al. [154] carried out a randomized study on male piglets (n = 32) to evaluate the effects of different amounts of dietary AA: DHA at a constant ratio on the concentrations and circadian rhythms of bone metabolism biomarkers. Piglets were randomly assigned to one of four dietary treatments for 15 days: (1) control formula; (2) control formula supplemented with 0.5:0.1 g AA: DHA per 100 g of fat; (3) control formula supplemented with 1.0:0.2 g AA: DHA/100 g of fat; and (4) control formula supplemented with 2:0.4 g AA:DHA per 100 g of fat. Supplements contained a constant AA:DHA ratio of 5:1 and a total ω-6:ω-3 ratio of 9:1. Two biomarkers were measured to determine bone turnover: plasma osteocalcin (OC), a non-collagenous protein secreted by osteoblasts and released into the circulation during bone fracture [155], was sampled at 09:00 h, 15:00 h, and 21:00 h on day 15 as an indicator of both bone formation and resorption; and N-telopeptide of type 1 collagen cross-links (NTx), a sensitive and specific marker of bone resorption [156], was measured in urine samples collected from 21:00 h on day 14 and 09:00 h, 09:00–15:00 h, and 15:00–21:00 h on day 15. There was a significant effect of diet on urinary on NTx, with the 1.0:0.2 g AA: DHA group presenting lower values of NTx: creatinine and, therefore, reduced bone resorption compared with the other groups. A significant effect of time on NTx concentrations was also found, with lower concentrations at 21:00 h (6 hrs. postprandial) compared with 09:00 h. (12 h postprandial) and 15:00 h (6 h postprandial). There was no time by diet interaction on NTx: creatinine, suggesting that dietary PUFAs did not affect the circadian rhythms of bone turnover markers. No significant effects of either diet or time were observed for plasma OC values, although the authors partly attributed these results to the fact that blood was sampled over a 12 period rather than over 24 h.

Taking into consideration the fact that the retina contains the highest levels of DHA [157], Ikemoto et al. [158] decided to explore the influence of ω-3 fatty acids on the activities and circadian rhythms of lysosomal enzymes in the retina and pineal glands of rodents. Male pups from female Donryu rats receiving either a safflower oil (ω-3-deficient)-supplemented diet (n = 9) or perilla oil (ω-3 rich)-supplemented diet (n = 9) were fed the same diet as their mothers for 6 weeks. The sacrifice took place at 05:00 h, 09:00 h, 13:00 h, and 17:00 h, and the following enzyme activities were measured in both the retina and the pineal gland: β-glucosidase, β-glucuronidase, hexosaminidase, and acid phosphatase. In the safflower oil group, reduced DHA concentrations were found in the retina (<40%) and the pineal gland (<16%), which were partially and completely compensated by an increase in AA levels, respectively. In the retina, lysosomal activities at 05:00 h (dark period) were decreased and gradually increased after light exposure. Significantly decreased activities of the four lysosomal enzymes were found in the retina in safflower oil-supplemented rats compared with the perilla oil-fed group. In the latter, β-glucuronidase and hexosaminidase activities reached a plateau at 09:00 h that was not observed in the former. Nevertheless, no significant differences between dietary groups were observed in the activities or circadian rhythms of lysosomal enzymes in the pineal gland. The results suggest that a DHA-deficient diet negatively impacts the retinal lysosomal enzyme activities and may reduce the turnover rate of rod outer segments, which are daily hydrolyzed by these enzymes after light exposure for further reutilization.

In the two same groups of rats, the aforementioned authors additionally assessed the effects of DHA deficiency on phospholipid synthetic enzymes and their circadian variations in the retina [159]. For this purpose, the activities of CTP:phosphocholine cytidyltransferase (CT), and CTP:phosphoethanolamine cytidyltransferase (ET), the rate-limiting enzymes in the de novo synthesis of phosphatidylcholine (PC) and phosphatidylethanolamine (PE), were evaluated in the retinas excised at 05:00 h, 09:00 h, 13:00 h, and 17:00 h. In the safflower oil group, significant decreases in the DHA component of both PC and PE were observed, together with a significant increase in the content of AA. CT and ET activities showed a significant diurnal rhythm in both the perilla oil- and the safflower oil-fed rats: activities were the lowest at 05:00 h, during the dark period, and then considerably increased after exposure to light. Although the circadian variation in enzymatic activities was not altered by an ω-3-deficient diet, activities of CT and ET were significantly lower in the safflower oil group compared with the perilla oil group at all time points: for CT activity, the difference between groups was markedly lower at 09:00 h (43% lower in the safflower oil group compared with the perilla oil group); however, differences in ET activity between dietary groups were relatively constant at each time point (15–20%).

The selected studies on the circadian regulation of metabolic, reproductive, cardiovascular, and biochemical functions by ω-3 fatty acids are shown in Table 4.

To summarize, Figure 1 illustrates the diverse physiological functions controlled by ω-3 fatty acids via clock genes.

## 6. Discussion

The present review aims at collecting evidence on the role played by ω-3 fatty acids as circadian rhythm synchronizers and non-photic zeitgebers, emphasizing the molecular aspects of the influence exerted by ω-3 fatty acids on the biological clock. The varied studies found highlight the diversity of physiological functions that are under circadian control and can be modified by ω-3 fatty acid levels through prior regulation of the molecular clock. For better comprehension, articles have been grouped according to the physiological function assessed, paying attention to the anti-inflammatory and cognitive actions derived from ω-3 fatty acids.

We found four animal studies [113,114,115,116] evaluating the influence of ω-3 fatty acids on melatonin secretion and the sleep/wake cycle as the most representative circadian rhythm [24]. According to Zaouali-Ajna et al. [115] and Lavialle et al. [116], reduced DHA content in pineal phospholipids and increased pineal ω-6:ω-3 ratio were associated with diminished nocturnal secretion of melatonin in the pineal gland. Changes in melatonin concentrations induced by fluctuations in ω-3 fatty acid levels could be due to disruptions in pathways involved in the synthesis [114] and release of melatonin, as also suggested by Gazzah et al. [113]. In contrast to Zaouali-Ajna et al. [115], Zhang et al. [114] observed changes in diurnal pineal melatonin concentrations in ω-3-deficient rats, finding an inverse relationship between DHA content in pineal phospholipids and daytime pineal melatonin levels. As Zaouali-Ajna et al. themselves suggested, the divergences from Zhang et al. [114] could be explained by the age difference of tested animals (12 months vs. 2 months), as ω-6/ω-3 and AA/DHA ratios experience substantial variations as the age increases. Therefore, considerably greater ratios might have allowed Zaouali-Ajna et al. [115] to observe significant differences in melatonin concentrations during the light phase.

The anti-inflammatory actions derived from the molecular clock regulation by ω-3 fatty acids were tested in four studies, two of them conducted on cell cultures [118,121] and the other two on murine models of NAFLD [122,127]. The counteracting effects of ω-3 fatty acids, particularly DHA, on inflammation may involve multiple mechanisms, such as PPAR inhibition, the JAK-STAT-SOC pathway, and T lymphocyte differentiation. According to these studies, the main targets of the molecular clock for ω-3 fatty acids were: RORα, through the MaR1-RORα-12-LOX autoregulatory circuit loop [127]; and reasonably the BMAL1 component of the primary TTFL [118,122], which is upregulated by RORα [23] and was previously reported to inhibit NF-kB activation and the release of proinflammatory cytokines [44]. Interestingly, upregulation of BMAL1 expression in mHypoE-37 neurons and hepatic cells by dietary patterns favoring inflammation (palmitate, Western-style diet) was also reported in two of the four studies [118,122]. Possible explanations for these results may be that phase-shifting effects induced by saturated fatty acids on the circadian oscillatory profile of BMAL1 [118] could inhibit its anti-inflammatory effects, or that the actions derived from BMAL1 on inflammation are cell-specific. BMAL1 deletion altered B and T cell migration to lymph nodes [48], and it also disrupted the rhythmic cycling of Ly6C^hi^ monocytes, predisposing mice to acute infections and chronic inflammatory diseases [43]. However, macrophages lacking BMAL1 showed increased motility and phagocytic function, conferring protection against pneumococcal pneumonia [42]. It may also be plausible that, beyond the cell types, the effects derived from BMAL1 deletion on the global immune defense depend on the type of immunity triggered by specific pathogens. Indeed, in other cells from the myeloid lineage, such as eosinophils, a more inflammatory phenotype derived from BMAL1 deletion induced more severe asthma attacks [58], as given by an exaggerated Th2-driven immune response.

Three studies were found on the regulation of neurological functions by ω-3 fatty acids via clock genes, two of them conducted on rodents [130,131] and one on patients with diverse neurodegenerative conditions [132]. According to Minami et al. [130], DHA-enriched diets were reported to synchronize locomotor activity with the light/dark period in rats. These findings were not reproduced by Umezawa et al. [131], although the type of animal tested and the fact that SHRSP usually shows lack of synchronization with the light/dark phases after the onset of stroke. In patients with MCI, Fiala et al. [132] reported that the addition of ω-3 supplements to cholinesterase inhibitors slowed the progression to dementia. These changes coincided with an increase in energy genes and amyloid β-phagocytosis in peripheral mononuclear cells in parallel with an upregulation in ARNTL2 expression and a trend towards an increase in CLOCK expression.

Divergences were observed between Mozaffarian et al. [151] and Buchhorn et al. [152] regarding the impact of dietary ω-3 fatty acid consumption on the circadian fluctuations of heart rate. These could have been due to the methodological differences between studies: recruited sample in terms of clinical diagnosis, age and medication intake, as well as the intervention (variable supplementation versus usual intake through food-frequency questionnaire and the corresponding recall bias).

The interaction between ω-3 fatty acids and the molecular clock should not only be understood as exerted by the former on the latter but as a bidirectional relationship. In murine models, urinary levels of fatty acid metabolites, such as 4-hydroxy-5E, 7Z, 10Z, 13Z, 16Z, and 19Z-docosahexaenoic acid (4-HDoHE), were higher in CLOCK mutant compared with wild-type mice [160]. From a neurological point of view, RORα deficiency was associated with alterations in fatty acid accretions in the cerebellum and secondary neurological impairments [161]. Plasma concentrations of DHA and EPA were reported to follow distinct circadian rhythmicity in healthy adults with normal habits of eating and sleeping [162]. In a convergent functional genomics-approached study on women with psychiatric disorders, circadian clock genes, particularly PER 1, were found to be one of the top biomarkers for suicidal ideation, with DHA signaling appearing as one of the most overrepresented biological pathways in these biomarkers [163].

We are aware that our research may have several limitations. Although it is focused on the molecular influence exerted by ω-3 fatty acids on the biological clock, not all the studies reported the effect on specific clock genes, but rather light/dark variations on measured biomarkers. Second, this is a comprehensive and extensive work covering a wide range of functions; for a few of these, only a couple of studies were found showing controversial results. However, we intended to provide enough detail to explain inconsistencies between studies. Third, this is a narrative review and, as such, our work is essentially informative and has limited capacity to assess result robustness and provide original evidence. Nonetheless, in an attempt to overcome this limitation, we have carefully collected evidence according to a detailed, a priori-designed search strategy and grouped studies assessing similar physiological functions to highlight consistencies and controversies between them.

## 7. Conclusions

According to existing literature, the molecular clock participates as an intermediary in the regulation exerted by ω-3 fatty acids on a variety of physiological functions, particularly inflammatory/immune and cognitive processes. This means that ω-3 fatty acids could serve as non-photic zeitgebers and prove therapeutically beneficial for circadian disruption-related pathologies. Future work is required to devise the specific effects of particular ω-3 fatty acids on individual clock genes and the reciprocal influence of the molecular clock over ω-3 fatty acid turnover.

## Figures and Tables

**Figure 1 ijms-23-12162-f001:**
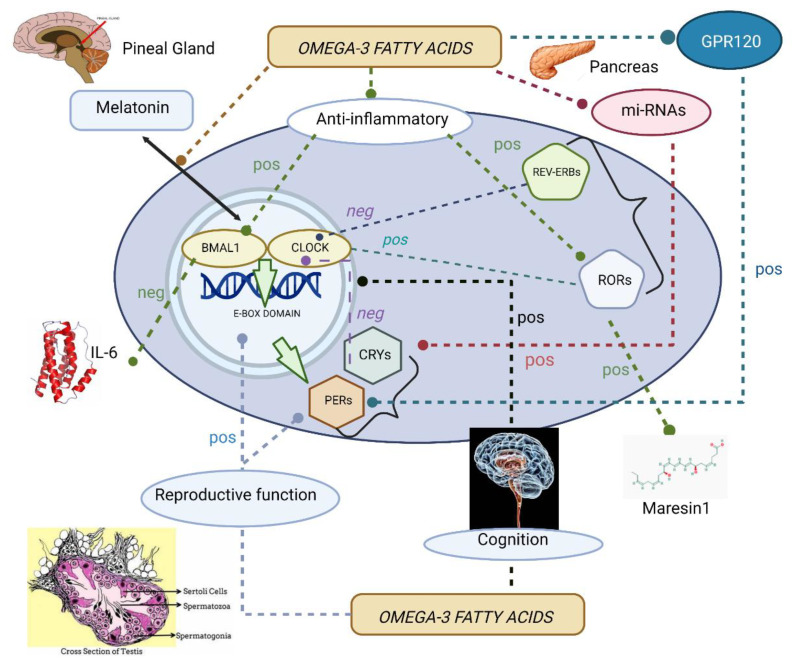
Schematic representation of the molecular machinery of the biological clock and modulation by omega-3 fatty acids to regulate the diversity of physiological functions. Abbreviations: GPR120 = G-protein coupled receptor 120; miRNAs = microribonucleic acids; pos = positive feedback; neg = negative feedback; IL-6 = interleukin-6; BMAL1 = brain and muscle Arnt-like protein 1; CLOCK = circadian locomotor output cycles kaput; PER = period; CRY = cryptochrome; REV-ERB = reverse erythroblastoma; ROR = retinoid-orphan receptor.

**Table 1 ijms-23-12162-t001:** Selected studies on the regulation of melatonin secretion by omega-3 fatty acids.

First Author	Year	Design	Population	Duration	Experimental Diet(s)	Control	Outcome *
Gazzah	1993	pCT	Rats	Not found	Peanut oil (ω-3 deficient)	Peanut plus rapeseed oil (ω-3 sufficient)	↓ DHA in pineal phospholipids ↓ Pineal melatonin release
Zhang	1998	pCT	Rats	12 m	26% LA + 0.06% ALA (ω-3 deficient)	12% LA + 1% ALA and DHA (ω-3 sufficient)	↓ DHA in pineal phospholipids ↓ Pineal 12-HETE ↑ Diurnal pineal melatonin levels
Zaouali-Ajna	1999	pCT	Rats	5 w	G1: Peanut oil (ω-3 deficient); G2: peanut oil plus DHA-enriched phospholipids	G3: Peanut plus rapeseed oil (ω-3 sufficient); G4: peanut plus rapeseed oil and DHA-enriched phospholipids	↑ Pineal ω-6:ω-3 ratio in G1 vs. G3 ↑ Pineal DHA and ω-3 fatty acids in G2 vs. G1 ↓ Nighttime urinary aMT6 concentrations in G1 vs. G3 ↑ Nighttime urinary aMT6 concentrations in G2 vs. G1 No significant differences in daytime urinary aMT6 concentrations between groups
Lavialle	2008	pCT	Hamsters	3 m	Peanut oil (ω-3 deficient)	Peanut plus rapeseed oil (ω-3 sufficient)	↑ AA:DHA in the striatum and pineal gland ↑ Locomotor activity ↑ DA, DOPAC and HVA in the striatum ↓ Nocturnal melatonin secretion peak

Pineal ω-6:ω-3 ratio influenced melatonin release/production, lypoxygenase metabolite concentrations, and the levels of dopamine and dopaminergic metabolites. * Outcomes were referred to the experimental group when there were two groups of comparison. Abbreviations: pCT = preclinical trial; m = months; w = weeks; ω-3 = omega-3; LA = linoleic acid; ALA = alpha-linolenic acid; DHA = docosahexaenoic acid; 12-HETE = 12-hydroxyeicosatetraeonic acid; ω-6 = omega-6; G1 = group 1; G2 = group 2; G3 = group 3; G4 = group 4; aMT6 = 6-sulfatoxymelatonin; DA = dopamine; DOPAC = 3,4-dihydroxyphenylacetic acid; HVA = homovalinic acid.

**Table 2 ijms-23-12162-t002:** Selected studies on the regulation of inflammatory pathways by omega-3 fatty acids via clock genes.

First Author	Year	Design	Population	Duration	Experimental Diet(s)	Control	Outcome *
Greco	2014	pCT	mHypoE-37 neurons	36 h	G1: treatment with SFA (palmitate); G2: treatment with ω-3 FAs (DHA); G3: pre-treatment with DHA followed by treatment with palmitate + DHA	None	↑ BMAL1/REV-ERBα expression in G1 vs. control The non-significant phase delay of BMAL1/REV-ERBα in G1 vs. control No changes in TLR4, IL-6, or AgRP expression in G1 ↑ BMAL1 expression in G2 vs. control ↑ BMAL1 period in G2 vs. control Significant phase-advance of BMAL1 in G2 vs. control ↓ IL-6 expression in G2 vs. control ↓ BMAL1 expression in G3 vs. G1 No phosphorylation changes of GSK3β/AMPK in G1, G2 or G3
Kim	2016	pCT	Bmal1-dLuc fibroblasts Adipocytes	48 h	G1: treatment with SFA (palmitate); G2: treatment with ω-3 FAs (DHA)	BSA	Large phase advances in G1 vs. control ↑ IL-6 and NK-kB expression in G1 vs. control: maximal effects coinciding with the largest cell-dependent phase shifts Small phase delays in G2 vs. control ↓ IL-6 expression in G2 vs. control: maximal effects coinciding with the largest cell-dependent phase shifts DHA and AICAR (AMPK activator) attenuated palmitate-induced inflammatory and circadian changes
Yuan	2016	RpCT	Rats	16 w	WD; FOH	Normal chow	↓ Serum TG, TCh, LDL-C, AST, ALT in FOH vs. WD ↓ Macrovesicular steatosis in FOH vs. WD ↓ IL-1β and TNF-α in FOH vs. WD ↓ Sreb1f, Socs 2, Ccl-2, and Sema4a in FOH vs. WD ↑ BMAL1 expression in WD: reversed by FOH ↓ PER2/PER3 expression in WD: reversed by FOH
Han	2019	pCT	Mice: WT and RORα-MKO	2 w	HFD + DHA	HFD	M2 polarity shift in liver macrophages in WT (not in RORα-MKO) ↑ RORα expression via MaR-1 in WT (not in RORα-MKO) ↓Liver injury markers in WT (not in RORα-MKO)

Anti-inflammatory changes induced by ω-3 fatty acid-enriched diet were related to modifications of BMAL1/REV-ERBα and RORα expression. * Outcomes were referred to the experimental group when there were two groups of comparison. Abbreviations: pCT = preclinical trial; RCT = randomized preclinical trial; h = hours; m = months; w = weeks; G1 = group 1; G2 = group 2; G3 = group 3; SFA = saturated fatty acids; ω-3 = omega-3 fatty acids; DHA: docosahexaenoic acid; BMAL1 = brain and muscle Arnt-like protein 1; REV-ERBα = reverse erythroblastoma α; TLR4 = toll-like receptor 4; IL-6 = interleukin 6; AgRP = Agouti-related protein; GSK3β = glycogen synthase kinase 3β; AMPK = adenosine monophosphate-activated protein kinase; FAs = fatty acids; NF-kβ = nuclear factor kappa β; WD = Western-style lard-rich diet; FOH = ω-3-enriched fish oil diet; TG = triacylglycerol; TCh = total cholesterol; LDL-C = low-density lipoprotein cholesterol; AST = aspartate aminotransferase; ALT = alanine aminotransferase; IL-1β = interleukin 1-beta; TNF-α = tumor necrosis factor alpha; Sreb1f = sterol regulatory element binding factor 1; Socs2 = suppressor of cytokine-signaling 2; Ccl-2 = chemokine (C-C motif) ligand 2; Sema4a = semaphoring 4a; PER2 = period 2; PER3 = period 3; WT = wild type; RORα-MKO = myeloid-specific RORα knockout; MaR-1 = maresin-1.

**Table 3 ijms-23-12162-t003:** Selected studies on the regulation of neurological functions by omega-3 fatty acids via clock genes.

First Author	Year	Design	Population	Duration	Experimental Diet(s)	Control	Outcome *
Minami	1997	RpCT	Rats (SHRSP)	14 w	G1: semipurified diet with 1% DHA; G2: semipurified diet with 5% DHA	Semipurified diet (Clea) with 0% DHA	↓ SBP in G1 and G2 vs. control (DHA dose-dependent manner) L/D desynchronization in ambulatory activity in control L/D synchronization in ambulatory activity in G1 and G2 ↓ Plasma and brain AA in G1 and G2 vs. control ↑ Plasma and brain DHA in G1 and G2 vs. control (DHA dose-dependent manner)
Umezawa	1999	pCT	Mice (SAMR1)	15 m	Safflower oil (ω-6:ω-3 = 250.3)	Perilla oil (ω-6:ω-3 = 0.24)	↑ Locomotor activity in the open-field test No differences between groups in L/D synchronization of locomotor activity ↑ Incorrect responses in L/D discrimination tests ↓ DHA, EPA, and total ω-3 PUFAs in brain PC/PE
Fiala	2020	CT	Neurodegenerative patients: SCI (*n* = 8); MCI (*n* = 8); AD (*n* = 6); FTD (*n* = 1); VCI (*n* = 2); DLB (n = 3)	54 m	CI/ST + Smartfish (DHA 5 mg/mL, EPA 5 mg/mL)	CI/ST	↓ Progression to dementia in MCI by 26 to 54 m (MMSE scores) ↑ Aβ-phagocytosis by PBMCs in MCI ↑ Energy genes in PBMCs in MCI: hexokinase, GAPDH, OGDH, NAD ↑ ARNTL2 expression in MCI Non-significant upregulation of CLOCK expression in MCI

Neurological actions derived from DHA in mice were associated with light/dark synchronization of locomotor activity and reduced incorrect responses in light/dark discrimination tests; in neurodegenerative patients, delayed progression to dementia was related to changes in the expression of CLOCK/ARNTL2. * Outcomes were referred to the experimental group when there were two groups of comparison. Abbreviations: RpCT=randomized preclinical trial; pCT = preclinical trial; SHRSP = stroke-prone spontaneously hypertensive rats; w = weeks; mo = months; DHA = docosahexaenoic acid; G1 = group 1; G2 = group 2; SBP = systolic blood pressure; L/D = light/dark; AA = arachidonic acid; SAMR1 = senescence-resistant strain 1; ω-6 = omega-6 fatty acids; ω-3 = omega-3 fatty acids; EPA = eicosapentaenoic acid; ω-3 PUFAs = omega-3 polyunsaturated fatty acids; PC = phosphatidylcholine; PE = phosphatidylethanolamine; SCI = subjective cognitive impairment; MCI = mild cognitive impairment; AD = Alzheimer’s disease; FTD = frontotemporal dementia; VCI = vascular cognitive impairment; DLB = dementia with Lewy bodies; CI = cholinesterase inhibitors; ST = other standard therapies; MMSE = Mini-Mental State Examination; PBMCs = peripheral blood mononuclear cells; Aβ-phagocytosis = amyloid beta-phagocytosis; GAPDH = glyceraldehyde 3-P dehydrogenase; OGDH = alpha-ketoglutarate; NAD = nicotinamide adenine dinucleotide; ARNTL2 = aryl hydrocarbon receptor nuclear translocator-like 2; CLOCK = circadian locomotor output cycles kaput.

**Table 4 ijms-23-12162-t004:** Selected studies on the regulation of metabolic, reproductive, cardiovascular, and biochemical functions by omega-3 fatty acids via clock genes.

Selected Studies on the Regulation of Metabolic Functions by Omega-3 Fatty Acids via Clock Genes							
First Author	Year	Design	Population	Duration	Experimental Diet(s)	Control	Outcome *
Daimiel-Ruiz	2015	pCT	Caco-2 cells	Not specified	G1: treatment with Ch; G2: treatment with DHA; G3: treatment with CLA	None	↑ miR-107 in all groups (only significant in G3) ↑ CLOCK expression after miR-107 inhibition ↑ PER2 expression after miR-107 overexpression ↓ CRY2 expression after miR-107 inhibition Changes in circadian oscillations of CLOCK, PER, and CRY after transfection with miR-107 mimic
Furutani	2015	pCT	Mice (RF)	3-4 d	G1: sardine oil; G2: menhaden oil; G3: tuna oil; G4: saury oil; G5: soybean oil with DHA/EPA	AIN-93M formula	Liver clock phase delay in all groups vs. control (larger in G3) Liver clock phase shifts similar between G3 and G5 ↑ Blood insulin levels in G3 and G5 ↑ Liver PER2 expression in G3 and G5 Changes in the liver clock, insulin levels, and PER2 by DHA/EPA in RF are mediated by GPR120/FFAR4
Amos	2019	pCT	Mice (Bob-cat: CAT overexpression)	8 w	ω-3-enriched diet	HFD	tSabilized glucose and insulin levels ↑ GPR120/FFAR4 expression ↑ Nfr2 expression CAT and HO-1 ↑ Preserved circadian patterns of food intake
**Selected studies on the regulation of reproductive functions by ω-3 fatty acids via clock genes**							
**First Author**	**Year**	**Design**	**Population**	**Duration**	**Experimental Diet(s)**	**Control(s)**	**Outcome ***
Wang	2018	RpCT	Mice (males)	12 w	G1: HFD (17.14% ω-6 FAs + 0.33% ω-3 FAs); G2: FO (10.98% ω-6 FAs + 10.96% ω-3 FAs)	Standard chow (53% ω-6 FAs + 1% ω-3 FAs)	↑ Body weight, perinephric fat weight, and perinephric/epididymal fat weight/body ratios in G1 and G2 vs. control ↓ Sperm and Leidyg cell nº in G1 vs. control Sperm and Leidyg cell nº similar between G2 and control ↓ Expression of LHR, STAR, p450scc, 3β-HSD, 17β-HSD, and SF-1 in G1 vs. control ↓ Testosterone/oestradiol ratio in G1 vs. control Expression of LHR, STAR, p450scc, 3β-HSD, and SF-1 restored in G2 vs. G1 ↑ Testosterone/oestradiol in G2 vs. G1 No differences in expression levels of LHR, STAR, p450scc, 3β-HSD, 17β-HSD, SF-1, BMAL1, CLOCK, CRY1, CRY2, and PER2 between ZT0 and ZT12 in control Remarkable differences in expression levels of LHR, STAR, p450scc, 3β-HSD, 17β-HSD, SF-1, BMAL1, CLOCK, CRY1, CRY2, and PER2 between ZT0 and ZT12: differences abolished in G2 ↓ BMAL1, CLOCK, CRY2, and PER2 expression in G1: changes reversed in G2
**Selected studies on the circadian regulation of cardiovascular functions by ω-3 fatty acids**							
**First Author**	**Year**	**Design**	**Population**	**Duration**	**Experimental Diet(s)**	**Control(s)**	**Outcome ***
Mozzafarian	2008	Observational (CS)	Men and women ≥ 65 years of age	N/A	N/A	N/A	Associations between ω-3 FA consumption and HRV measures: -Positive associations between ω-3 FA intake and measures reflecting vagally mediated respiratory variation -Inverse associations between ω-3 FA intake and erratic sinoatrial firing -No significant associations between ω-3 FA intake and circadian variation of HRV
Buchhorn	2018	CT (pre-post)	ADHD children and adolescents	143.7 d (average)	ω-3 fatty acid-enriched supplements + ST (post)	ST (pre)	Associations between ω-3 FA intake and HR/HRV measures: -Inverse association between ω-3 FA and HR -Positive association between ω-3 FA and HRV -Significant effect of ω-3 FA on HR and HRV circadian variation measures: MESOR of mean HR, rMSSD, and HF; the amplitude of mean HR and rMSSD
Bhardwaj	2021	RCT	CHD and HT patients	24 w	G1: CHD patients receiving BFO (high quantities of ALA); G2: HT patients receiving BFO	CHD/HT patients receiving SO	↓ MESOR of mean HR, SBP, and DBP in G1 and G2 ↓ The amplitude of SBP and DBP in G1 and G2 Significant differences in the acrophase of SBP/DBP in G2
**Selected studies on the circadian regulation of biochemical processes by ω-3 fatty acids**							
**First Author**	**Year**	**Design**	**Population**	**Duration**	**Experimental Diet(s)**	**Control(s)**	**Outcome***
Mollard	2008	RpCT	Piglets (male)	15 d	G1: formula supplemented with 0.5:0.1g AA:DHA/100 g of fat; G2: formula supplemented with 1.0:0.2 g AA:DHA/100 g of fat; G3: formula supplemented with 2.0:0.4 g AA:DHA/100 g of fat	Formula with AA:DHA=0	↓ Bone resorption (NTx:creatinine) in G2 vs. other groups ↓ Urinary NTx levels at 21:00 h vs. 09:00 h and 15:00 h No effect of AA:DHA on NTx circadian variation No effect of diet or time on plasma OC levels
Ikemoto	2000	pCT	Rats	6 w	Safflower oil (ω-3-deficient diet)	Perilla oil (ω-3-enriched diet)	↓ DHA levels in the retina and PG ↓ Activities of lysosomal enzymes (β-glucosidase, β-glucuronidase, hexosaminidase and acid phosphatase) in the retina β-glucuronidase and hexosaminidase activities reached a plateau at 09:00 h in the retina No differences in activities or circadian rhythms of lysosomal enzymes in the PG
Ikemoto	2000	pCT	Rats	6 w	Safflower oil (ω-3-deficient diet)	Perilla oil (ω-3-enriched diet)	↓ DHA content in PC and PE in the retina ↓ CT and ET activities in the retina No differences in circadian variation of CT and ET activities in the retina

Interactions between omega-3 fatty acids and clock genes on the regulation of metabolic functions are mediated by microRNAs and GPR120/FFAR4. Changes in the dietary composition of omega-3 fatty acids influenced reproductive maturation and function in male mice via changes in the expression of BMAL1, CLOCK, PER2, and CRY2. Heartrate variability is influenced by dietary consumption of omega-3 fatty acids, although the impact on HRV measures differed between studies. Diverse biochemical processes are affected by the proportion of omega-3 fatty acids in the diet, such as bone resorption and lysosomal activities in the retina. * Outcomes were referred to the experimental group when there were two groups of comparison. Abbreviations: pCT = clinical trial; RpCT = randomised preclinical trial; Ch = cholesterol; G1 = group 1; G2 = group 2; G3 = group 3; G4 = group 4; G5 = group 5; DHA = docosahexaenoic acid; CLA = conjugated linoleic acid; miR-107 = microribonucleic acid-107; CLOCK = circadian locomotor output cycles kaput; PER2 = period 2; CRY2 = cryptochrome 2; d = days; RF = restricted feeding; EPA = eicosapentaenoic acid; GPR120/FFAR4 = G-protein coupled receptor 120/free fatty acid receptor 4; CAT = catalase; ω-3 = omega-3 fatty acids; HFD = high-fat diet; Nrf2 = nuclear factor E2-related factor 2; HO-1 = heme oxygenase; ω-6 = omega-6 fatty acids; FAs = fatty acids; FO = fish oil-enriched diet; nº = number; LHR = luteinizing hormone receptor; STAR = steroidogenic acute regulatory receptor; p450scc = cholesterol side-chain cleavage enzyme; 3β-HSD = 3β-hydroxysteroid dehydrogenase; 17β-HSD = 17β-hydroxysteroid dehydrogenase; SF-1 = steroidogenic factor 1; BMAL1 = brain and muscle Arnt-like protein 1; CLOCK = circadian locomotor output cycles kaput; CRY1 = cryptochrome 1; CRY2 = cryptochrome 2; PER2 = period 2; ZT0 = zeitgeber 0 (morning); ZT12 = zeitgeber 12 (night); CS = cross-sectional; HRV = heart-rate variability; 24-h rMSSD = average of daytime and nighttime parasympathetic respiratory variation; ADHD = attention-deficit and/or hyperactivity disorder; ST = standard therapy; HR = heart rate; MESOR = rhythm adjusted 24-h mean level; amplitude = distance between MESOR and the highest maximum value of the cosine curve; HF = high-frequency HRV via fast Fourier transformation; CHD = coronary heart disease; HT = essential hypertension; BFO = blended flaxseed oil; ALA = alpha-linolenic acid; SO = sunflower oil; SBP = systolic blood pressure; DBP = diastolic blood pressure; acrophase = highest values in 24-h day; AA = arachidonic acid; OC = osteocalcin; NTx = N-telopeptide of type 1 collagen cross-links; PG = pineal gland; PC = phosphatidylcholine; PE = phosphatidylethanolamine; CT = CTP:phosphocholine cytidyltransferase; ET = CTP:phosphoethanolamine cytidyltransferase.

## Data Availability

Not applicable.

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
