# Peer review of "Role of Omega-3 Fatty Acids as Non-Photic Zeitgebers and Circadian Clock Synchronizers"

_ijms, 2022, doi:10.3390/ijms232012162_

Round 1
Reviewer 1 Report
A narrative review exploring the relationship between omega-3 fatty acid diets and circadian biological physiology.
The authors made an evident effort and work to collect information from a variety of physiological and pathological studies, with no limitations, including cell cultures, animal models, and human studies.
The result is a collection of studies organized by common physiological functions. The review is dense reading but informative.
There is a clear limitation in the number of studies included in each section. However, the total collection of evidence supports the relationship between omega-3 fatty acid diets (as a no photonic regulator) and the circadian biological clock.
No major comments or refusal of the manuscript is recommended by this reviewer.
Minor proposals for the authors' consideration (not required a new review by this author):
- Line 92 focuses on the last three actions.
- Discussion. Discuss the controversial data (e.g., exposed in some introduction data, references 40 and 56) concerning BMAL1, due is relevance in many studies. Discuss the lack of experimental studies versus cell studies in some points, such as 3. /inflammation.
- Help readers follow the review, by reducing some collection of evidence and adding some definitions or explanations (there are several fields of knowledge involved in the review), such as ketosis diet, or flaxseed diet content of 3-FA, versus sunflower.
- English review. The language writing is well done, but the last review is recommended.
- Caution with some conclusion, consider rewriting the sentence, such as 471-473
- Table; outcome results of the omega-3 fatty diet groups. Detailed in captions.
- Section 5. is highly recommended for a better exploration of the review goal, however, maybe there are many sub-sections (such as bone metabolism); the 5.3 can be summarized.
- Material and methods detail the exclusion criteria used to eliminate articles based on the reading of the abstract and titles.
- Suggest to the authors: deep in the discussion (including limitations)
- Note: where is page 33 of 43?
Reviewer 2 Report
This article reviews the involvement of omega-3 fatty acids as synchronizers of circadian clock. The authors have compiled a large amount of information, which perhaps in some cases is excessive and goes beyond the scope of the review. Although there are aspects that can be better organized, the review may be interesting for readers. The areas for improvement are discussed below.
1. In an introduction about the biological clock, one cannot speak only of the Light-entrainable oscillator (LEO). Food-entrainable oscillator (FEO) must be included, more so if we take into account that the omega-3 fatty acids would be a clear entry with food, possibly with a more relevant role on FEO than on LEO. In fact, authors refer to omega-3 fatty acids as non-photic zeitgebers.
2. The review is focused on human studies and animal models. It would be convenient in the abstract or in the introduction to specify the animal models used.
3. The justification and aim of this review (section 1.4., page 6) could be introduced at the beginning so that readers have a clear idea of the review they are going to read. If you don't have to wait for 6 pages of an overly long introduction, you may lose interest in the most interesting aspects of this review.
4. The tables summarize very clearly the information explained in the manuscript. In any case, perhaps an integrative scheme would help to understand the role of omega-3 fatty acids as circadian clock synchronizers.
